# Unifying Diarization, Separation, and ASR with Multi-Speaker Encoder

## Abstract

The rapid progress of single-task architectures has dominated recent developments in multi-talker speech processing, prompting the need for unified approaches. This paper introduces a unified multi-speaker encoder (UME), a novel model architecture that jointly learns representations for diarization, separation, and multi-speaker automatic speech recognition (ASR) tasks using a shared pre-trained foundational speech encoder. We leverage the hidden representations from multiple layers of UME to effectively use information from different semantic levels, contributing to bottom-up alignment between tasks. This joint training approach captures the inherent interdependencies among the tasks, enhancing overall performance on overlapping speech data. Our evaluations demonstrate that UME achieves substantial improvements over the single-task state-of-the-art (SOTA) baselines dedicated to speaker diarization, speech separation, and multi-speaker ASR. Notably, for speaker diarization, UME achieved SOTA performance by lowering the diarization error rate (DER) from 3.24 to 2.19 on the Libri2Mix dataset. Furthermore, our results in multi-speaker ASR outperform the previous results, reducing the concatenated minimum-permutation word error rate (cpWER) from 11.9 to 9.2 on the LibriSpeech2Mix evaluation set.

## 1 Introduction

Speaker diarization (SD), speech separation (SS), and multi-speaker automatic speech recognition (ASR) are tasks of great importance that aim to comprehend and answer the question "who spoke what and when," with applications to transcribing meetings and interviews, among others. Previous studies in SD (Fujita et al., 2019a;b; Horiguchi et al., 2022), SS (Luo & Mesgarani, 2019; Wang et al., 2023), and multi-speaker ASR (Qian et al., 2018; Seki et al., 2018; Chang et al., 2020) have focused primarily on improving the quality of single-task models that operate independently on acoustic information to separate or label speaker segments and transcribe the text in a speech-processing system (Watanabe et al., 2020; Chen et al., 2020; Raj et al., 2021a). A key limitation of training tasks independently is that inter-dependencies cannot be leveraged.

Most existing frameworks address this limitation by unifying speech-processing architectures (Boeddeker et al., 2024; Kalda et al., 2024). These architectures consist of either a joint ASR/SD (Mao et al., 2020), SS/ASR (Kanda et al., 2022), or a SD/SS (Maiti et al., 2023) task following a fixed optimal order that can vary depending on the target scene scenario (Watanabe et al., 2020; Chen et al., 2020; Raj et al., 2021a). These different target scenes suggest that we solve these tasks jointly, independent of the order, so all these tasks can benefit from each other.

Lately, there has been a shift towards employing pre-trained speech foundation models (SFMs) (Chen et al., 2022; Radford et al., 2023; Peng et al., 2024a) in end-to-end (E2E) systems, which effectively learn useful representations for various speech processing tasks (Yang et al., 2021). However they do not work well on multi-speaker conversation recognition. Additionally, it has been demonstrated that different layers encode different types of information in SD and ASR tasks (Chen et al., 2022). Preliminary observation from these studies shows that intermediate layers of the encoder extract a rich hierarchy of information, e.g., in WavLM large, initial layers and last layers are more critical for SD and ASR tasks. Therefore, it makes sense to utilize multiple layers to jointly optimize all SD, SS, and ASR tasks effectively. The question, therefore, naturally arises: *can we build a unified model that leverages all encoder layers to optimize performance across multiple tasks?*

Motivated by the potential of SFMs and E2E speech processing, we propose a unified multi-speaker encoder (UME), a novel E2E speech-processing framework. The proposed framework is generalizable to use any SFM, E2E SD, SS and multi-speaker ASR task. We selected OWSMv3.1 (Peng et al., 2024b) as the shared encoder for this framework due to its widespread recognition, reproducibility, open-source availability, and fast, efficient encoding capabilities. EEND (Horiguchi et al., 2022) as a SD model due to its efficient overlapped E2E speech processing and Conv-TasNet (Luo & Mesgarani, 2019) as a SS method as it is a very well-known separation model for time-domain overlapped speech handling and multi-speaker ASR (Chang et al., 2020) for its superior speech recognition performance in the E2E overlapped speech recognition. UME jointly optimizes all these tasks into a single network with multitask learning to minimize the error accumulation for a speech processing framework. Additionally, by extracting features from all the layers of the OWSMv3.1 shared encoder, we can effectively learn better-hidden representations from the encoder layers, bringing better information exchange and bottom-up alignment to all the tasks from different semantic levels. We argue that such an E2E framework should provide a shared representation space for SD, SS and multi-speaker ASR tasks and preferably have strong generalizability and learnability.

We conduct extensive experiments on different design choices of UME using typically complete overlapped speech from the Libri2Mix dataset. The contributions are summarized as follows:

- We propose a unified speech-processing framework to jointly optimize the performance of SD, SS, and multi-speaker ASR tasks with hidden representations of the speech foundation encoder.

- We propose using a weighted sum of the pre-trained speech foundation encoder layers to simplify the connection between each task.

- We demonstrate the effectiveness of our framework on two-speaker and three-speaker overlapped speech and obtain substantial performance improvement in each diarization, separation, and multi-speaker ASR task.

## 2 RELATED WORK

### 2.1 MULTI-LAYER FEATURE LEARNING

Multi-layer feature learning has been explored as an efficient approach for fully leveraging the information present in various layers of neural networks to enhance the representation and generalization abilities in single task for speech processing (Yang et al., 2021; Chen et al., 2022), natural language processing (Peters et al., 2018; Dou et al., 2018), and computer vision (Zheng et al., 2021; Naseer et al., 2021). In the field of computer vision, researchers (Zheng et al., 2021; Naseer et al., 2021) improved semantic segmentation performance by aggregating features from different layers of visual transformers, whereas in natural language processing, a weighted sum (Peters et al., 2018) of representations from intermediate RNN layers or aggregation (Dou et al., 2018) of attention layers was explored as an input for different task heads. Similar ideas have also been explored in single-task speech processing models (Yang et al., 2021; Chen et al., 2022) to analyze the effect and contribution of intermediate layers on single downstream task performance. However, the existing weighted sum of hidden representations from different layers of SFM is not explicitly explored for multiple task heads in a unified speech model, and there has been no exploration into their suitability for the end-to-end speech processing framework.

### 2.2 JOINT TRAINING

Joint training (Watanabe et al., 2017) approaches have achieved significant success in speech processing. With the rapid advancements in multitask learning-based joint training methods, researchers combined tasks like SS and SD within a single neural network (Neumann et al., 2019; Kinoshita et al., 2020), using various task combinations. Previous works have focused on joint training of pairs such as ASR/SD (Shafey et al., 2019; Mao et al., 2020), SS/ASR (Kanda et al., 2022), or SS/SD (Maiti et al., 2023; Boeddeker et al., 2024). Our work represents the first effort to jointly train the SD, SS, and multi-speaker ASR within one unified model so that all tasks can benefit from each other.

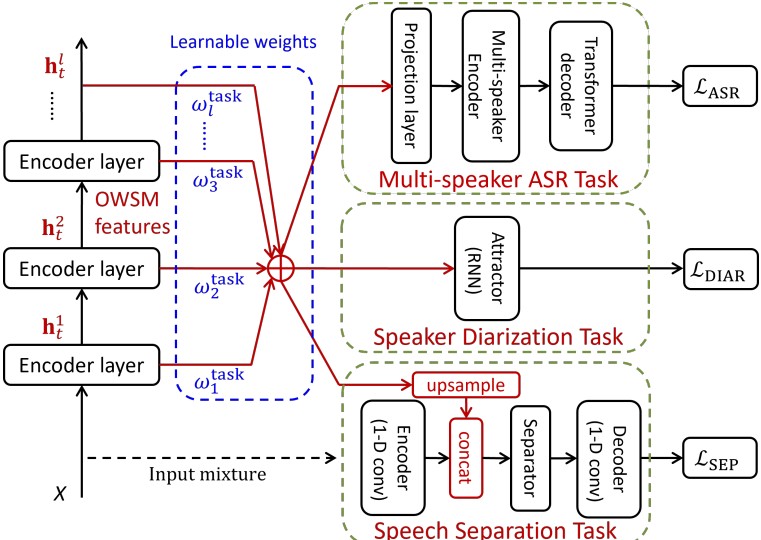

Figure 1: Illustration of UME framework.

To address this gap, we build UME to assess the suitability of SD, SS, and multi-speaker ASR tasks for constructing the unified E2E speech processing framework. To the best of our knowledge, this is the first work that utilizes an explicit pre-trained SFM in an E2E way for all three tasks, i.e., SD, SS, and multi-speaker ASR, while also leveraging the hidden representations and allowing the flow of information through a weighted sum of intermediate layers.

## 3 UNIFIED MULTI-SPEAKER ENCODER (UME)

Figure 1 shows the overall framework of UME, which leverages the hidden representations through a weighted sum of intermediate layers that act as the bridge between SD, SS, and multi-speaker ASR tasks to enable comprehensive and detailed interaction from each layer of the SFM encoder. Our goal is not to develop new encoder or speech processing tasks; in principle, one can apply any SFM encoder, SD, SS, or multi-speaker ASR tasks in the proposed speech processing framework.

### 3.1 INPUT SPEECH MIXTURE

We start with the $T$-length single-channel input speech mixture $X = \{x_t \in \mathbb{R} | t = 1, \cdots, T\}$ of $C$ speakers. We define the input speech mixture in an anechoic condition given by:

$$x_t = \sum_{c=1}^{C} y_{(c,t)} s_{(c,t)} + n_t, \tag{1}$$

where, $s_{(c,t)} \in \mathbb{R} | t = 1, \cdots, T$ is the $T$-length source speech signal of speaker $c$, $n_t \in \mathbb{R} | t = 1, \cdots, T$ is the noise signal and $y_{c,t} \in \mathbb{R} | t = 1, \cdots, T$ is the speech activity of speaker $c$ indicating that $y_{(c,t)} = 1$ if speaker $c$ is talking at time $t$ and otherwise. This creates a ground truth speaker label sequence $Y = \{y_{(c,t)} \in \{0,1\}^C | t = 1, \cdots, T'\}$ for the SD task in Section 3.3 and will be estimated as $\hat{Y} = \{\hat{y}_{(c,t)} \in \{0,1\}^C | t = 1, \cdots, T'\}$ where a $T$-length input speech signal is subsampled to $T'$-length after the feature extraction in Section 3.2.

### 3.2 SPEECH FOUNDATION MODEL ENCODER

Recent works (Peng et al., 2024b) show that OWSM's encoder has strong and efficient encoding capabilities on various downstream tasks. We can note that OWSM was trained on single-speaker speech-to-text tasks (i.e., no speaker tasks in pre-training). But we can still adapt it to our multi-speaker setup. The speech encoder is a stack of $N$ E-Branchformer (Kim et al., 2023) encoder layers

that transforms the $T$-length single-channel input speech mixture $X = \{\mathbf{x}_t \in \mathbb{R}|t = 1, \cdots, T\}$ into a $D'$-dimensional subsampled $T'(< T)$-length hidden state representations $H_{(l)} = \{\mathbf{h}_t \in \mathbb{R}^{D'}|t = 1, \cdots, T'\}$ of $C$ speakers, where $l$ is a layer index from 1 to $N$. The simplified speech encoder can be represented as:

$$H_{(l)} = \text{SpeechEnc}_{\text{mix}}(X),\tag{2}$$

For inclusion in the task specific models with in a joint network, all layers of the $H_{(l)}$ are arranged in a single feature vector. Similar to (Yang et al., 2021), we compute a task-specific weighting of all the intermediate layers.

$$H = \sum_{l=1}^{N} \omega_{(l)}^{\text{task}} H_{(l)}.\tag{3}$$

In equation 3, $\omega_{(l)}^{\text{task}}$ are softmax-normalized learnable weights that scale the hidden state representations from different encoder layers to aid the optimization process for all the tasks during training.

## 3.3 SPEAKER DIARIZATION TASK

Given the robust performance of EEND (Horiguchi et al., 2022) with permutation invariant training (PIT) in estimating multi-speaker activities within an E2E framework, we adopt EEND for the SD task in the proposed UME E2E speech processing framework. The SD involves predicting speaker activity as binary multi-class labels by estimating the speaker label sequence $\hat{Y} = \{\hat{y}_{(c,t)} \in \{0,1\}^C|t = 1, \cdots, T'\}$, where $\hat{y}_{(c,t)} = 1$ indicates that speaker $c$ is active at time $t$, and $\hat{y}_{c,t} = 0$ otherwise. Unlike clustering-based methods (Bullock et al., 2020; Raj et al., 2021b), which often show poor performance in scenarios with simultaneous speaker activity, as they rely on distinct clusters that do not account for temporal overlap, however, the E2E approach can effectively model overlapped speech by explicitly setting $\hat{y}_{(c_1,t)} = 1$ and $\hat{y}_{(c_2,t)} = 1$ when both speakers are active at the same time.

Given the encoded hidden state representations $H_{(l)}$ from the speech encoder (Section 3.2) we map the speaker activity probabilities $\mathbf{p}_t \in \{0,1\}^C$ using a linear layer and an element wise sigmoid function $\sigma(\cdot)$, i.e.,

$$\mathbf{p}_t = \sigma(W\mathbf{h}_t + b).\tag{4}$$

where $W$ and $b$ are trainable weights and biases of the fully connected layer. We train EEND with PIT using speaker activity probabilities and the target speaker activity labels. The binary cross entropy-based (BCE) diarization loss ($\mathcal{L}_{\text{diar}}$) is optimized for all set of possible permutations.

$$\mathcal{L}_{\text{diar}} = \min_{\phi \in \Phi(C)} \sum \text{BCE}(\mathbf{y}_t^\phi, \mathbf{p}_t).\tag{5}$$

where $\Phi(C)$ contains a set of all possible speaker permutations $C$ and vector $\mathbf{y}_t^\phi$ contains the permuted reference of speaker labels.

## 3.4 SPEECH SEPARATION TASK

Speech separation is the task of predicting the separated speech signals $\hat{s}_1, \cdots \hat{s}_C \in \mathbb{R}|t = 1, \cdots, T$ for $C$ number of speakers for a given input speech mixture (Section 3.1). Since Conv-TasNet (Luo & Mesgarani, 2019) is a well-known time-domain speech separation architecture, we adopt Conv-TasNet as our speech separation task. It predicts the separated speech signals of $C$ speakers using a fully convolutional encoder, separation, and decoder network. The input speech mixture is first encoded through a 1-D convolutional encoder, resulting in $M$-dimensional hidden state representations $H'_{(l)} = \{\mathbf{h'}_t \in \mathbb{R}^M|t = 1, \cdots, T\}$.

$$H'_{(l)} = \text{ConvEnc}(X).\tag{6}$$

To take full advantage of the pre-trained OWSMv3.1 speech encoder and increase the resolution of the speech separation task, we concatenate the upsampled weighted sum of hidden state representations ($H_{(l)}$) extracted in Section 3.2 of the speech encoder with the encoded features $H'_{(l)}$ of

the Conv-TasNet in the separator network at the last layer ($l = N$) where they are further processed by a repeated stack of 1-D dilated temporal convolutional networks (TCNs) and extracts an $E$-dimensional embeddings $\mathbf{e}_t \in \mathrm{R}^E$:

$$H_{(l)}^{\text{concat}} = \text{TCNs}(\text{Concat}(H'_{(l)}, \text{upsample}(H_{(l)}))), \tag{7}$$

$$\mathbf{e}_t = \text{TCNs}(\text{Conv}_{1\times 1}(\text{LayerNorm}(H_{(l)}^{\text{concat}}))). \tag{8}$$

The separation network then estimates the masks $\mathbf{m}_{(c,t)} \in [0,1]^M$ in equation 9 and computes the representation for each source $\mathbf{d}_{(c,t)} \in \mathrm{R}^M$ using element wise multiplication $\odot$ in equation 10.

$$\mathbf{m}_{(c,t)} = \sigma(\text{Conv}_{1\times 1}(\text{PReLU}(\mathbf{e}_t))), \tag{9}$$

$$\mathbf{d}_{(c,t)} = \mathbf{h}_t^{\text{concat}} \odot \mathbf{m}_{(c,t)}. \tag{10}$$

Finally, the Decoder recovers the separated audio signals $\hat{s}_{(c,t)}$ using a 1-D transposed convolutional layer.

$$\hat{s}_{(c,t)} = \text{Decoder}(\mathbf{d}_{(c,t)}). \tag{11}$$

The separation task is trained with the $\text{SI} - \text{SDR}$ loss ($\mathcal{L}_{\text{sep}}$) as defined below:

$$\mathcal{L}_{\text{sep}} = -10\log_{10} \frac{\left\| \frac{\langle \hat{s}_c, s_c \rangle s_c}{\|s_c\|^2} \right\|}{\left\| \hat{s}_c - \frac{\langle \hat{s}_c, s_c \rangle s_c}{s_c} \right\|^2} \tag{12}$$

### 3.5 Multi-speaker ASR Task

The multi-speaker ASR task, as adopted from (Chang et al., 2020), extends a joint connectionist temporal classification (CTC)/attention-based framework to recognize speech from multiple speakers within an E2E neural network. In the UME architecture, the input hidden state representations (Section 3.2) from the speech encoder are first encoded. Subsequently, each speaker's speech is extracted through $J$ speaker-differentiating encoder blocks (SpeakerEnc$_{\text{SD}}$). These speaker-dependent features are then transformed into $D''$-dimensional subsampled $T''(< T')$-length hidden state representations $H_{(l)}^j = \{\mathbf{h}_t^j \in \mathbb{R}^{D''} | t = 1, \cdots, T''\}$, where $j = \{1, \cdots, J\}$ for each speaker.

$$H_{(l)}^j = \text{SpeakerEnc}_{\text{SD}}^j(H). \tag{13}$$

The attention-based decoder generates the $U$-length output sequence $Y_{(l)}^j = \{y_u^j \in \mathcal{V} | u = 1, \cdots, U\}$, where $y_u$ is an output token at position $u$ in the vocabulary $\mathcal{V}$ for speakers $j = 1, \cdots, J$. PIT (Section 3.3) is employed to control the reference sequences $Y_{(l)}^j$ permutation. Specifically, PIT is applied to the CTC loss ($\mathcal{L}_{\text{ctc}}$) immediately after the encoder.

$$\hat{\pi} = \arg \min_{\pi \in \mathcal{P}} \sum_{j=1}^{J} \mathcal{L}_{\text{ctc}}(y_u^{\hat{\pi}(j)}, H_{(l)}^j). \tag{14}$$

where $\mathcal{P}$ is the set of all perumtaions on speakers $1, \cdots, J$, and $\hat{\pi}(j)$ is the $j$-th element of perumutation $\pi$. This ensures that the model is invariant to the order of the speaker labels, enhancing its ability to recognize and differentiate between multiple speakers accurately. Finally, the loss for the multi-speaker ASR ($\mathcal{L}_{\text{asr}}$) task is optimized using CTC and cross-entropy loss of the attention decoder ($\mathcal{L}_{\text{att}}$):

$$\mathcal{L}_{\text{asr}} = \lambda_{\text{ctc}}\mathcal{L}_{\text{ctc}}(y_u^{\hat{\pi}(j)}, H_{(l)}^j) + (1 - \lambda_{\text{ctc}})\mathcal{L}_{\text{att}}(y_u^{\hat{\pi}(j)}, H_{(l)}^j). \tag{15}$$

### 3.6 Training Objective

The UME framework optimizes all the three tasks using a multi-task learning loss function.

$$\mathcal{L}_{\text{all}} = \lambda_{\text{diar}}\mathcal{L}_{\text{diar}} + \lambda_{\text{sep}}\mathcal{L}_{\text{sep}} + \lambda_{\text{asr}}\mathcal{L}_{\text{asr}}. \tag{16}$$

The loss function is a weighted sum of $\mathcal{L}_{\text{diar}}$ in equation 5, $\mathcal{L}_{\text{sep}}$ in equation 12 and $\mathcal{L}_{\text{asr}}$ in equation 15. $\lambda_{\text{diar}}, \lambda_{\text{sep}}$, and $\lambda_{\text{asr}}$ are the weighting hyperparameters which are optimized empirically.

## 4 EXPERIMENTS

### 4.1 DATASET

In UME, we aim to optimize all three tasks: diarization, separation, and multi-speaker ASR, using multi-task learning in a unified framework. We require three ground truths to objectively evaluate performance, i.e., diarization labels, separated sources, and text for each speaker. While real-world multiparty datasets (Carletta et al., 2006; Horiguchi et al., 2021; Kamo et al., 2024) exist for diarization-only tasks, they often need separated sources and text. Therefore, we employ simulated conversation-like open-source datasets for training and evaluation. For training, we utilized the LibriMix (Cosentino et al., 2020) dataset. For evaluation, we employed both LibriMix and LibriSpeech-Mix (Kanda et al., 2020) datasets. LibriMix is a simulated dataset that generates speech mixtures using samples from LibriSpeech (train-clean100/train-clean360/dev-clean/test-clean) (Panayotov et al., 2015) and noise samples from WHAM! (Wichern et al., 2019). This dataset includes training, validation, and testing sets for two-speaker (Libri2Mix) and three-speaker (Libri3Mix) mixtures. This study reports results solely on Libri2Mix (two-speaker) and Libri3Mix (three-speaker) to effectively manage computational resources and reduce carbon footprint. This work used a 16kHz sampling rate, the "mixboth (i.e., includes speaker mixtures and WHAM noise)" method, and the "max" mode with 100% overlap. We choose the "max" mode as the ASR task is unfeasible compared to the "min" mode due to the truncation of speech signals on minimum-length sequences. We evaluated our system using the Libri2Mix and Libri3Mix datasets with a complete 100% overlap, as well as the LibriSpeech2Mix and LibriSpeech3Mix datasets, which include a partial random overlap of at least 0.5 seconds. The minimum, maximum, and average durations of the utterances for the training and evaluation sets are shown in Table 1. Additionally, a more comprehensive analysis of the characteristics of the training and evaluation sets is provided in Appendix A.2.

Table 1: The minimum, maximum, and average durations of utterances in the training and evaluation sets reported in seconds (s).

| Datasets | Minimum (s) | Maximum (s) | Average (s) |
|---|---|---|---|
| Libri2Mix - training set | 3.12 | 29.73 | 14.55 |
| Libri3Mix - training set | 4.21 | 29.74 | 15.13 |
| Libri2Mix - test set | 3.08 | 21.26 | 8.41 |
| Libri3Mix - test set | 3.23 | 20.91 | 9.00 |
| LibriSpeech2Mix - test set | 2.58 | 51.26 | 11.98 |
| LibriSpeech3Mix - test set | 3.32 | 56.77 | 16.23 |

### 4.2 EVALUATION METRICS

We evaluate diarization performance using the diarization error rate (DER%) (Fiscus et al., 2006) with a collar tolerance of 0.0 seconds and median filtering applied over 11 frames. For the separation task, we evaluate using five objective metrics and report the results in source-to-distortion ratio improvement (SDRi (dB)) (Vincent et al., 2006), scale-invariant source-to-distortion ratio improvement (SI-SDRi (dB)) (Roux et al., 2019), scale-invariant signal-to-noise ratio improvement (SI-SNR), short-time objective intelligibility (STOI) (Taal et al., 2010), and signal to artifacts ratio (SAR) Vincent et al. (2006) along with signal-to-interference ratio (SIR) (Vincent et al., 2006). We report the multi-speaker ASR performance using the concatenated minimum-permutation word error rate (cpWER) metric following prior work on permutation invariant training-based (PIT) multi-speaker ASR methods (Chang et al., 2020; Kanda et al., 2020). It involves selecting the lowest word error rate (WER) from the concatenated utterances of permuted speaker references and hypothesis files. Unlike the method in (Watanabe et al., 2020), our cpWER computation is independent of the speaker diarization branch, similar to (Chang et al., 2020), ensuring minimum error accumulation in the multi-speaker ASR process.

### 4.3 IMPLEMENTATION DETAILS

UME employs a pre-trained supervised SFM encoder, OWSMv3.1 (Peng et al., 2024b) medium, the feature extractor for all three tasks. We simplify the integration of the tasks following the evaluation in SUPERB (Yang et al., 2021) and WavLM (Chen et al., 2022) and provide learnable weights (Section 3.2) as input so that all the layers of the OWSMv3.1 contribute to the optimization of the tasks. Firstly, for the EEND (Section 3.3) task, we directly input the weighted sum of the extracted features with a frame length of 400 and a frameshift of 640 samples to the 1-layer RNN-based attractor with a hidden size of 1024. The EEND task thereby has an input-output dimension of 1024. Secondly, for the separation task (Section 3.4), we concatenate the 1024-dimensional hidden representations of the OWSMv3.1 with the 256-dimensional encoded features of the 1-D convolutional encoder in Conv-TasNet (Luo & Mesgarani, 2019). Since the OWSMv3.1 has a downsampling rate of 40ms, introducing a mismatch in the time dimension, we upsample the pre-trained representations for each time step to increase the resolution and ease the concatenation process. Following the concatenation, we input the 1280-dimensional concatenated features into the stack of three TCN blocks with eight convolutional layers with a hidden states dimension 512 for mask estimation. Finally, using a linear projection layer, we project the 1024-dimensional OWSMv3.1 features for the multi-speaker ASR task (Section 3.5) to 128 dimensions. We then introduce a Transformer-based post-encoder and decoder (Chang et al., 2020) with four speaker-differentiating encoder blocks (SpeakerEnc$_{SD}$) and six decoder blocks, each having 2048 linear units with an input dimension of 256. Before the post-encoder, we encode the OWSMv3.1 features by a convolutional layer with a subsampling factor of four. During training in UME, we initialize the encoder parameters with the pre-trained OWSMv3.1 medium encoder and fine-tune the encoder layers for 70 epochs, while all task-specific parameters have a flat start (i.e., no parameter initialization for task-specific layers) and are trained for 70 epochs. For the ASR-initialized UME version, the multi-speaker ASR model is pre-trained separately for 30 epochs, and then the ASR-specific head in the UME model is initialized from this pre-trained model. This results in a total of 70 epochs of fine-tuning for the OWSMv3.1 encoder layers, 70 epochs of training for the diarization and separation tasks, and 70 epochs of fine-tuning for the ASR task. We use the AdamW optimizer (Loshchilov & Hutter, 2019) with an initial learning rate of $4e − 4$ (optimized empirically) and weight decay $1e − 06$. The learning rate is warmed up for 10,000 steps and then decayed linearly to zero for the rest of the training steps. Four A100 80GB GPUs are used during training, and the batch size is dynamically adjusted based on the input length using the numel batch type in the ESPnet toolkit (Watanabe et al., 2018). In our experiments, the average batch size was 44, and it took six days to train the model for up to 70 epochs. For task-specific weights, we adopted a weighted-sum scalarization (Ehrgott, 2000) approach to simplify the multi-objective optimization problem into a single-objective (Bazgan et al., 2022) one by assigning equal weights to all task-specific losses (Section 3.6) (i.e., $\lambda_{asr} = 0.33$, $\lambda_{diar} = 0.33$, $\lambda_{sep} = 0.34$). This approach assumes that the tasks are cooperative rather than conflicting, particularly in our two-speaker and three-speaker scenarios, and reflects their equal importance in our framework. Furthermore, we explored a two-stage strategy to optimize task-specific weights, inspired by the 4D ASR work by (Sudo et al., 2023). However, as discussed in Section 5.2, this strategy resulted in performance degradation for one or more tasks. Since the primary goal of this study is to develop a unified framework capable of integrating multiple tasks rather than optimizing individual task performance, we propose an equal-weighting strategy that assigns equal importance to all tasks. This approach is validated by experimental results in Section 5.2, which demonstrate that simple equally weighted scalarization achieves state-of-the-art performance.

## 5 MAIN RESULTS

Tables 2, 4, and 3 show the performance of UME compared with previous works on downstream single task frameworks. With only 460 hours of simulated input speech mixture for training, UME achieves state-of-the-art performance, particularly 2.19% of DER on a 100% overlap Libri2Mix evaluation set, outperforming the previous state-of-the-art (SOTA) model WavLM (Chen et al., 2022). A similar trend of improvement also occurs for the SS and multi-speaker ASR tasks, which achieve the best performance. In this work, we also compare our results and report the findings by explicitly setting the multi-task learning weights of the individual tasks to zero in our unified framework for an unbiased comparison, providing more insights about the flexibility of our proposed method. In the following sections, we discuss the experimental results in detail.

Table 2: DERs (%) for two-speaker and three-speaker evaluations. No collar tolerance was allowed. **Bold:** the proposed method outperforms the baseline. **Underlined:** the best result.

| Method | Model | Libri2Mix | Libri3Mix |
|---|---|---|---|
| EEND (Horiguchi et al., 2022) | *Self-supervised pretrained* (Yang et al., 2021) | | - |
| | HuBERT Large (Hsu et al., 2021) | 5.75 | - |
| | wav2vec 2.0 Large (Baevski et al., 2020) | 5.62 | - |
| | WavLM Large (Chen et al., 2022) | 3.24 | - |
| | Other models in SUPERB(Yang et al., 2021) | 6.59-10.54 | - |
| EEND (Horiguchi et al., 2022) | *Without weighted sum* | | |
| | Reproduced | 4.62 | - |
| | UME ($\lambda_{\text{diar}} = 1.0$) | **2.91** | **3.26** |
| | UME ($\lambda_{\text{asr}} = 0.1, \lambda_{\text{diar}} = 0.1, \lambda_{\text{sep}} = 0.8$) | **2.28** | (diverged) |
| | *With weighted sum* | | |
| | UME ($\lambda_{\text{asr}} = 0.33, \lambda_{\text{diar}} = 0.33, \lambda_{\text{sep}} = 0.34$) | **2.26** | (diverged) |
| | + ASR initialized | **2.45** | **3.15** |
| | UME ($\lambda_{\text{asr}} = 0.1, \lambda_{\text{diar}} = 0.1, \lambda_{\text{sep}} = 0.8$) | **2.19** | (diverged) |
| | + ASR initialized | - | **2.84** |

## 5.1 END-TO-END SPEAKER DIARIZATION RESULTS

Table 2 shows that the most impressive result for the UME is SD, which outperforms WavLM (Chen et al., 2022) by 32% relatively in a 100% overlapped task setting for Libri2Mix. Furthermore, UME also achieved state-of-the-art results on Libri3Mix. Notably, WavLM is trained using overlapped speech mixtures, whereas OWSMv3.1 (Peng et al., 2024b) is trained solely on clean speech. Despite this, OWSMv3.1, adapted as the multi-speaker encoder having an improved architecture, outperforms WavLM. We hypothesize that the additional training losses from SS and multi-speaker ASR tasks provide additional granularity during the training in the multi-task learning framework. We verified this hypothesis by conducting an ablation study that explicitly set the weights of the SS ($\lambda_{\text{sep}}$) and multi-speaker ASR ($\lambda_{\text{asr}}$) losses to zero (i.e., $\lambda_{\text{diar}} = 1$) (Section 3.6), resulting in a substantial performance drop of the UME for the SD task. This indicates that the SS and multi-speaker ASR tasks enhance performance in the overlapped speech task.

Table 3: Since PIT does not enforce a fixed speaker order, the results are presented using cpWER (↓) for multi-speaker ASR on Libri2Mix, LibriSpeech2Mix, Libri3Mix and LibriSpeech3Mix evaluation sets. **Bold:** the proposed method outperforms the baseline. **Underlined:** the best result.

| Model | Libri2Mix | LibriSpeech2Mix | Libri3Mix | LibriSpeech3Mix |
|---|---|---|---|---|
| *Without weighted sum* | | | | |
| Multi-speaker Transformer (Chang et al., 2020) (reproduced) | 29.7 | 16.2 | - | - |
| + Speed perturbation (reproduced) | 24.4 | 12.7 | - | - |
| PIT LSTM-AED (Kanda et al., 2020) | - | 11.9 | - | 52.3 |
| SOT (Kanda et al., 2020) | - | 11.2 | - | 24.0 |
| UME ($\lambda_{\text{asr}} = 1.0$) | 25.0 | 13.0 | **26.4** | **16.0** |
| UME ($\lambda_{\text{asr}} = 0.33, \lambda_{\text{diar}} = 0.33, \lambda_{\text{sep}} = 0.34$) | **22.7** | **11.0** | (diverged) | (diverged) |
| + ASR initialized | **21.1** | **9.2** | 26.5 | **15.7** |
| UME ($\lambda_{\text{asr}} = 0.1, \lambda_{\text{diar}} = 0.1, \lambda_{\text{sep}} = 0.8$) | **22.4** | 11.9 | (diverged) | (diverged) |
| + ASR initialized | - | - | 27.3 | 20.3 |
| *With weighted sum* | | | | |
| UME ($\lambda_{\text{asr}} = 0.1, \lambda_{\text{diar}} = 0.1, \lambda_{\text{sep}} = 0.8$) | 25.5 | 12.8 | (diverged) | (diverged) |

## 5.2 MULTI-SPEAKER ASR RESULTS

For the multi-speaker ASR task, we input the OWSMv3.1 extracted features through a shallow speaker-differentiating encoder trained with CTC, attention, and PIT losses without using the SD and SS tasks. Similar to a previous study (Chang et al., 2020) which is our reproducible baseline, we initialized the SpeakerEnc$_{\text{SD}}$ blocks (Section 3.5) with a pre-trained model from the ESPnet recipe for training stability. For the multi-speaker ASR task in UME, we observe that the initialization of the ASR model provides training stability and outperforms the strong baselines in Table

$3^1$ both for 100% overlap (Libri2Mix & Libri3Mix) and partial overlap task (LibriSpeech2Mix & LibriSpeech3Mix). The LibriSpeech2Mix evaluation results demonstrate that initializing the ASR model leads to a 22.7% relative cpWER improvement compared to the PIT-based LSTM-AED (Kanda et al., 2020) method and a 17.9% relative cpWER improvement compared to the SOT-based (Kanda et al., 2020) approach, highlighting the robustness of our UME framework for multi-speaker speech recognition tasks. Moreover, the LibriSpeech3Mix evaluation results showed a 34.6% relative cpWER improvement compared to the SOT-based method. Our experiments further indicate that initializing the three-speaker model with a pre-trained two-speaker model is essential, as training the three-speaker model without such initialization consistently resulted in divergence. Notably, the UME framework was trained using the "mixboth" Libri2Mix and Libri3Mix training set, which combines two-speaker and three-speaker mixtures with WHAM noise, but was evaluated on the LibriSpeech2Mix and LibriSpeech3Mix evaluation set containing only clean speech. This demonstrates its superior generalization ability across datasets with varying data modeling characteristics. We also find that using a weighted sum of hidden state representations for multi-speaker ASR tasks results in performance degradation, as discussed in the following Section 5.4.

## 5.3 END-TO-END SPEECH SEPARATION RESULTS

Unlike previous studies (Yang et al., 2021; Chen et al., 2022) which report the separation results in "min mode", UME requires overlapped mixtures in "max mode" during the training process due to the unification of the ASR task, as discussed in Section 4.1. For this reason, we evaluate the UME on 100% overlapped mixtures in "max mode" with our fully reproduced Conv-TasNet model following the similar setup in Section 4.3 without the concatenated features. Experimental results in Table 4 show that the improvement for SS task is not as substantial compared with the SD (Table 2) and multi-speaker ASR (Table 3) tasks. However, we see a consistent improvement compared to the separation-only tasks in Table 4, indicating that concatenating encoded features with the upsampled hidden representations of the OWSMv3.1 encoder in the TCN block (Section 3.4) improves separation performance, resulting in an improved performance in the overall speech processing framework. An example of the effect of the concatenated features for two-speaker case in the separation task is shown in Figure 2, illustrating that the separation task shows improved performance in recovering the speaker activity without using an additional diarization branch. We also provide recovered speech examples for the three-speaker case in Appendix A.3.

Table 4: Speech separation results on the evaluation sets of Libri2Mix and Libri3Mix, using the "max mode" setting. The metrics STOI, SAR, SDR, SIR, and SI-SNRi are used to evaluate speech separation performance, with all values reported in decibels (dB). **Bold:** the proposed method outperforms the baseline. **Underlined:** the best result.

| Model | Libri2Mix | | | | | Libri3Mix | | | | |
|---|---|---|---|---|---|---|---|---|---|---|
| | STOI | SAR | SDR | SIR | SI-SNRi | STOI | SAR | SDR | SIR | SI-SNRi |
| *Without weighted sum* | | | | | | | | | | |
| ConvTasNet (Luo & Mesgarani, 2019) | 87.63 | 11.80 | 11.48 | 25.88 | 10.93 | - | - | - | - | - |
| UME ($\lambda_{sep} = 1.0$) | 89.13 | 12.60 | 12.39 | 27.67 | **11.81** | 85.31 | 10.61 | 10.16 | 22.45 | 9.53 |
| UME ($\lambda_{asr} = 0.1, \lambda_{diar} = 0.1, \lambda_{sep} = 0.8$) | **90.49** | **13.34** | **13.18** | **29.39** | **12.64** | | | (diverged) | | |
| *With weighted sum* | | | | | | | | | | |
| UME ($\lambda_{asr} = 0.33, \lambda_{diar} = 0.33, \lambda_{sep} = 0.34$) | **90.29** | **13.22** | **13.05** | **29.11** | **12.51** | | | (diverged) | | |
| + ASR initialized | **89.82** | **12.88** | **12.68** | **28.48** | **12.12** | 86.48 | 11.05 | 10.69 | 23.43 | 10.07 |
| UME ($\lambda_{asr} = 0.1, \lambda_{diar} = 0.1, \lambda_{sep} = 0.8$) | **90.82** | **13.55** | **13.39** | **29.70** | **12.84** | | | (diverged) | | |

## 5.4 EFFECT OF LAYER WEIGHTS

For the unification of SD (Section 3.3), SS (Section 3.4), and multi-speaker ASR (Section 3.5) tasks, we weighted sum the hidden state representations of the intermediate layers of the OWSMv3.1 encoder as an input for each task. Experimental results in Table 2 and Table 4 show that weighted sum representations improve the SD and SS performance while degrading the multi-speaker ASR task in Table 3. From the layer weights shown in Appendix A.1, we observe that the initial and final layers

---

[1]We excluded the results reported by (Meng et al., 2024), despite their better WER performance, because they trained their model on the "clean" subsets of Libri2Mix and Libri3Mix, while our study used the noisy "mix both" subset (Section 4.1).

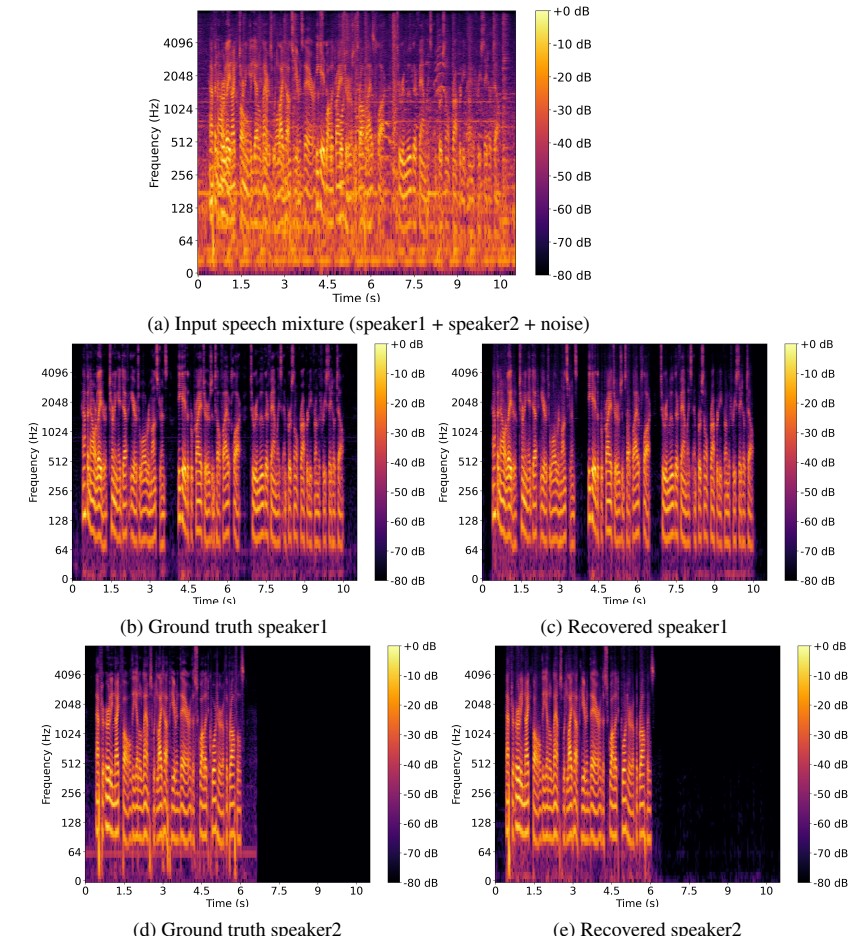

(a) Input speech mixture (speaker1 + speaker2 + noise)

(b) Ground truth speaker1

(c) Recovered speaker1

(d) Ground truth speaker2

(e) Recovered speaker2

Figure 2: An example of the effect of concatenation with OWSMv3.1 features on separated signals in UME. (a) Input speech mixture of two speakers and WHAM noise (speaker1, speaker2 and noise) with 100% overlap. (column 1) Ground truth for separated signals. (column 2) Recovered speech signals using separation branch output (after concatenation)

generally contribute more for all these tasks, obtaining higher weights. One possible explanation is that the parallel branch architecture (Peng et al., 2022) of the OWSMv3.1 encoder is effective at combining local and global information, giving higher weight to the top and bottom layers. As local signal information is necessary for speech reconstruction tasks, the SD and SS tasks must completely exploit the information contained in all the intermediate layers for speech reconstruction. However, sequence-to-sequence tasks like ASR, which requires long-term dependencies to learn contextually relevant features in the attention mechanism (Vaswani et al., 2017), perform poorly on weighted sum features as the maximum path length between any two input and output positions in networks composed of the different layers may be distorted by averaging the layer weights. This finding is consistent with the existing studies (Yang et al., 2021; Chen et al., 2022) on downstream tasks using single-task self-supervised speech frameworks.

# 6 CONCLUSION

In this paper, we propose UME, a unified framework for end-to-end speech processing, which integrates speaker diarization, speech separation, and multi-speaker ASR with a weighted sum of hidden states of the intermediate layers. UME substantially outperforms strong baselines and previous works and achieves state-of-the-art performance on speaker diarization task. In the future, we are interested in evaluating the performance on real datasets and extending it to a multilingual UME.

# 7 LIMITATIONS

While the proposed UME leveraging the weighted sum of hidden state representations of the intermediate layers as the bridge between speaker diarization, speech separation, and multi-speaker ASR tasks simultaneously achieving substantial improvements over previous works, it still has some limitations: (1) the absence of ground truth for all three tasks in real-world data makes it challenging to objectively evaluate the UME framework's performance. Therefore, we focus on simulated datasets, where ground truth is available, to ensure accurate comparisons. We acknowledge this limitation and plan to explore real-world datasets in future work as they become more accessible and standardized. (2) the current method employs a supervised pre-trained encoder, trained on clean and non-overlapped speech with a low time resolution of 40ms and shows suboptimal performance on the separation task; (3) the proposed UME only supports two-speaker and three-speaker tasks, and it would be nice to able to support unlimited number of speaker tasks; (4) we have to pre-train an independent model for speaker-differentiating heads to get optimal multi-speaker ASR performance due to small amount of simulated dataset in the current method, which is time and resource inefficient; (5) the effectiveness of applying UME to other speech domains (e.g., child speech, dialect speech) needs further investigation.

# 8 ETHICS STATEMENT

This work presents UME, utilizing an open-source pre-trained speech foundation model encoder OWSMv3.1 for unifying speaker diarization, speech separation, and multi-speaker ASR tasks. We implement our models using an open-source ESPnet framework. We evaluate our methods on standard benchmarks provided by the open-source research community. The datasets used in this study contain LibriMix, LibriSpeechMix extracted from LibriSpeech. They are all public datasets and are widely used in the research community. In preparing this manuscript, first author used generative AI tools, specifically ChatGPT, to assist in rewriting, rephrasing, and checking the grammar of certain sections. These tools were employed selectively to enhance the clarity and readability of the content, ensuring that the manuscript meets the highest standards of academic communication. All intellectual contributions, research findings, and interpretations remain entirely our own.

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

# A APPENDIX

## A.1 EFFECT OF LAYER WEIGHTS

Figure 3 illustrates a detailed analysis of the weight distributions observed under various training configurations, comparing joint training and single-task setups.

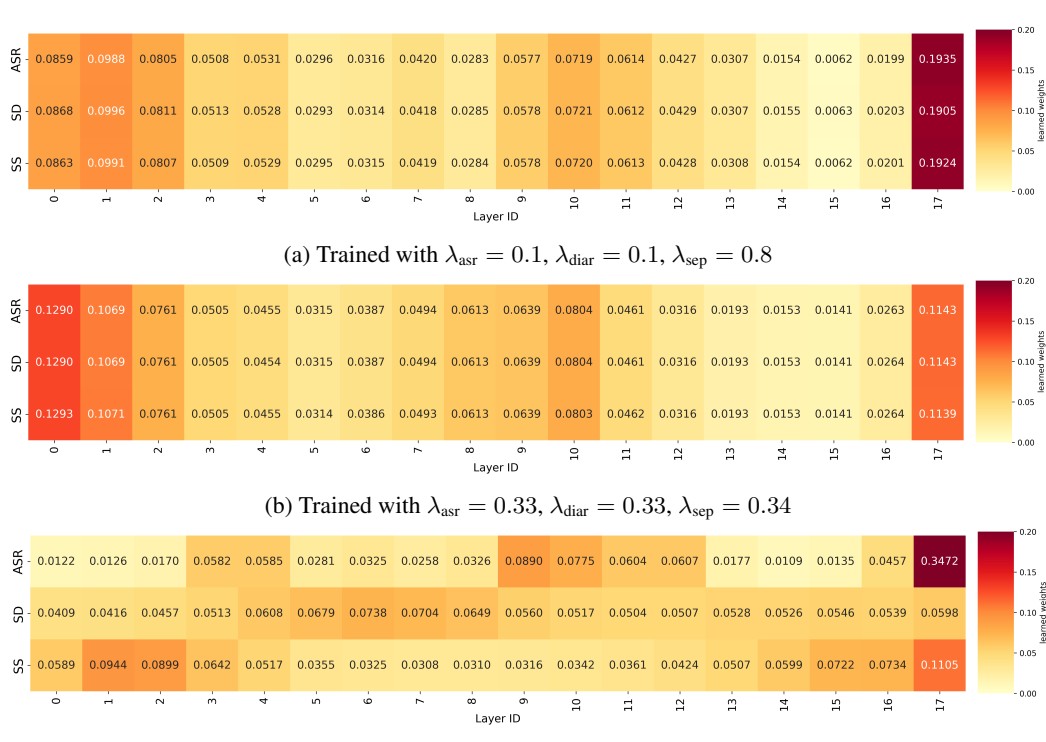

(a) Trained with $\lambda_{asr} = 0.1$, $\lambda_{diar} = 0.1$, $\lambda_{sep} = 0.8$

(b) Trained with $\lambda_{asr} = 0.33$, $\lambda_{diar} = 0.33$, $\lambda_{sep} = 0.34$

(c) Single-task models trained independently

Figure 3: Weight analysis for different training configurations.

## A.2 AUDIO DURATIONS

Figure 4 presents a comprehensive overview of the characteristics of the training and evaluation datasets used in this study. It provides insights into statistics such as the minimum, maximum, and average durations of utterances, as well as the total number of examples in each dataset, as discussed in Section 4.1.

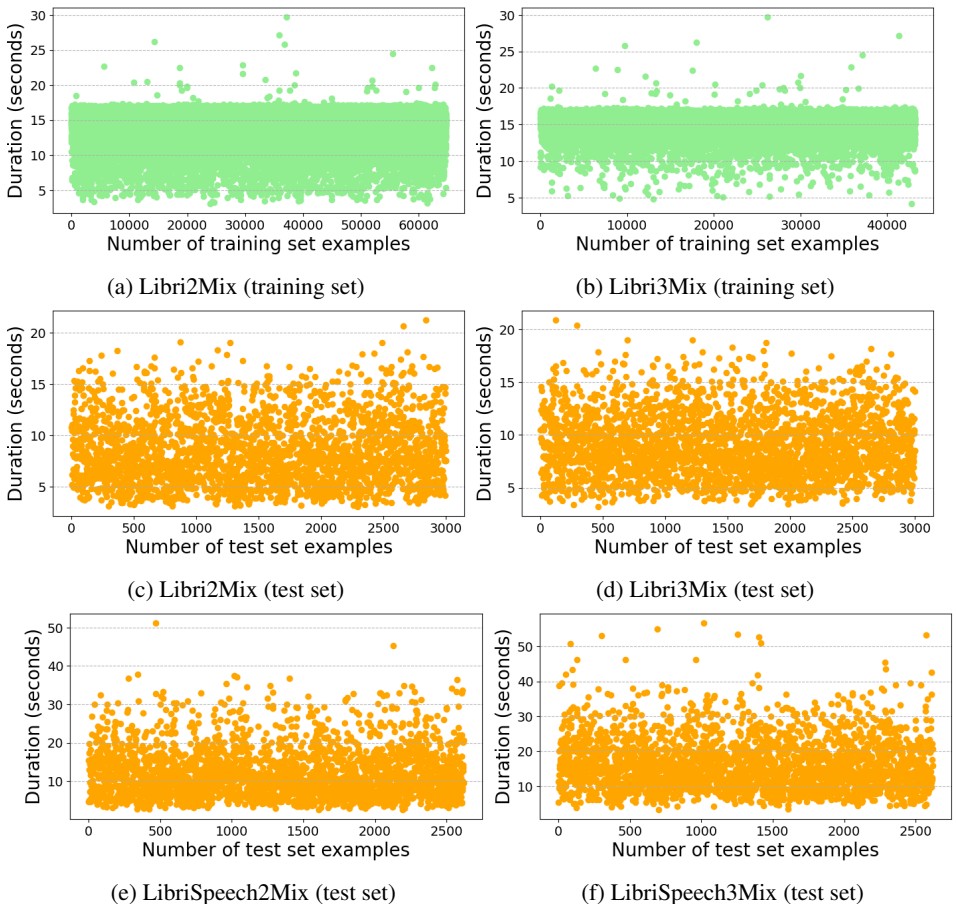

Figure 4: The number of examples in the training and evaluation sets, providing a comprehensive analysis of utterance durations in each set.

## A.3 SPEECH SEPARATION RESULTS FOR THREE SPEAKERS

Figure 5 presents the results of recovered speech obtained using the separation branch of the UME framework when trained on a three-speaker scenario (Libri3Mix dataset). The figure provides a detailed evaluation of the framework's performance, demonstrating its ability to accurately separate and recover individual speaker signals from a noisy speech mixture. These findings underline the practical applicability of the UME framework in real-world multi-speaker speech processing tasks.

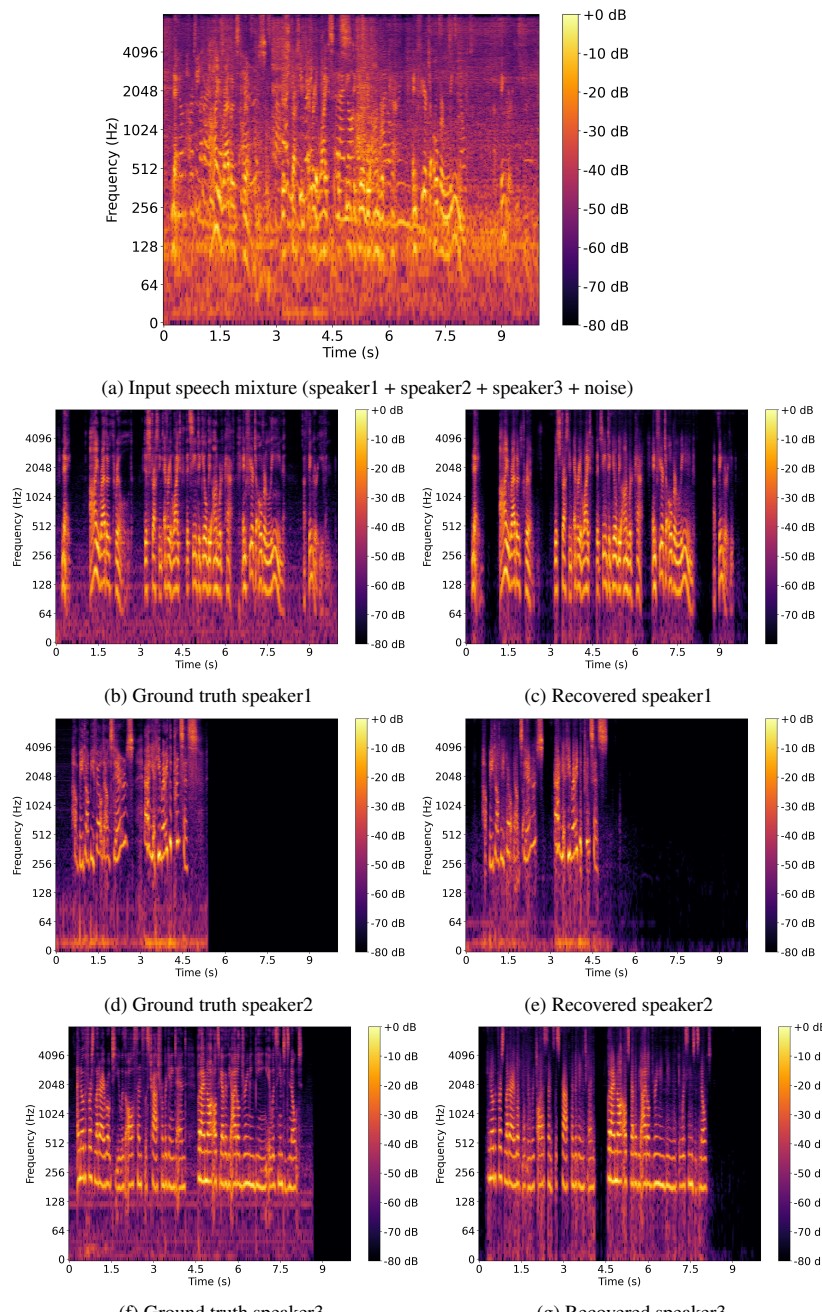

(a) Input speech mixture (speaker1 + speaker2 + speaker3 + noise)

(b) Ground truth speaker1           (c) Recovered speaker1

(d) Ground truth speaker2           (e) Recovered speaker2

(f) Ground truth speaker3           (g) Recovered speaker3

Figure 5: An example of the effect of concatenation with OWSMv3.1 features on separated signals in UME. (a) Input speech mixture of three speakers and WHAM noise (speaker1, speaker2, speaker3 and noise) with 100% overlap. (column 1) Ground truth for separated signals. (column 2) Recovered speech signals using separation branch output (after concatenation)

