# OpenReview forum: "Unifying Diarization, Separation, and ASR with Multi-Speaker Encoder"
_ICLR.cc/2025/Conference — Submitted to ICLR 2025_

### Official Review · Reviewer_wjfQ · 2024-10-30

**Soundness:** 2
**Presentation:** 2
**Contribution:** 2
**Rating:** 5
**Confidence:** 4

**Summary:**

This paper presents a method of combining diarization, separation and ASR tasks with multi-layer feature learning in a unifed model. Experiments have been conducted to evaluate the effectiveness of the proposed method.

**Strengths:**

The paper is written clearly and easy to follow.

**Weaknesses:**

1) The novelty is limited. As introduced in Section 2, multi-layer feature learning and joint training of speech tasks have been investigated in plenty of previous studies. The main contribution of this paper is to include more speech tasks into joint training and to combine it with multi-layer feature learning, which is limited in my opinion.
2) As showin in experimental results, the task weights for joint training in Eq.(16) play an important role in model performance. Thus, how to optimize these weights should be carefully explained.

**Questions:**

According to the results shown in Table 3 and 4, the layer weights for the three tasks were quite similar in both joint training configurations. Therefore, it may not be necessary to use task-dependent layer weights. How about the performance of using unifed layer weights across different tasks?

---

> ### Author Response · Authors · 2024-11-23
>
> **We thank the reviewer for the detailed feedback and constructive questions. We are encouraged by the recognition of the strengths of our work.**
>
> Below are the detailed responses to each query.
>
> **Weaknesses**
> >The novelty is limited. As introduced in Section 2, multi-layer feature learning and joint training of speech tasks have been investigated in plenty of previous studies. The main contribution of this paper is to include more speech tasks into joint training and to combine it with multi-layer feature learning, which is limited in my opinion.
>
> We acknowledge the reviewer’s comment regarding the lack of novelty in the multi-layer feature fusion/weighted sum approach. As noted in Section 2.1, our contribution lies in leveraging the weighted sum of OWSM encoder features to develop a unified speech model capable of addressing diarization, separation, and multi-speaker ASR tasks simultaneously. Integrating these tasks into a single multi-task framework presents a significant challenge due to their inherently disjoint nature. Our work addresses this issue by developing a unified framework that aligns these tasks through shared representations while allowing task-specific adaptations. This approach not only reduces redundancy in task-specific models but also promotes mutual benefits, such as improved speaker attributed transcription and better signal separation for diarization.
> ***
>
> > As shown in experimental results, the task weights for joint training in Eq.(16) play an important role in model performance. Thus, how to optimize these weights should be carefully explained.
>
> We adopted a weighted-sum scalarization approach [1] to simplify the multi-objective optimization problem into a single-objective [2] one by assigning equal weights to all task-specific losses. This approach assumes that the tasks are cooperative rather than conflicting, particularly in our two-speaker and three-speaker scenarios, and reflects their equal importance in our framework. Since the primary goal of this study is to develop a unified framework capable of integrating multiple tasks rather than optimizing individual task performance, we propose an equal-weighting strategy that assigns equal importance to all tasks. This approach is validated by experimental results, which demonstrate that simple equally weighted scalarization achieves state-of-the-art performance.
>
> We have added the above discussion in Section 4.3.
>
> ***
>
> **Questions**
>
> >According to the results shown in Table 3 and 4, the layer weights for the three tasks were quite similar in both joint training configurations. Therefore, it may not be necessary to use task-dependent layer weights. How about the performance of using unifed layer weights across different tasks?
>
> We will attempt to do the experiments with the unified layer weights. However, fully completing the experiments might not be feasible within the rebuttal timeline. We will do our best to include preliminary results during the revision phase.
>
> **References**
>
> ```
> [1] Matthias Ehrgott. Weighted Sum Scalarization, pp. 55–75. Springer Berlin Heidelberg, Berlin,Heidelberg, 2000. ISBN 978-3-662-22199-0.
> [2] Cristina Bazgan, Stefan Ruzika, Clemens Thielen, and Daniel Vanderpooten. The power of the weighted sum scalarization for approximating multiobjective optimization problems. Theory of Computing Systems, 66(1):395–415, Feb 2022. ISSN 1433-0490. doi: 10.1007/s00224-021-10066-5.
> ```

---

### Official Review · Reviewer_7DgA · 2024-10-30

**Soundness:** 2
**Presentation:** 2
**Contribution:** 1
**Rating:** 3
**Confidence:** 5

**Summary:**

The authors proposed a "Unified Multi-Speaker Encoder (UME)" for tasks such as speech separation, speech diarization, and multi-speaker ASR. Specifically, the UME employs a pre-trained OWSM encoder and is jointly fine-tuned across all three tasks. The latent embeddings from multiple layers of the UME are combined using a weighted sum with learnable coefficients. However, the novelty of this approach is limited, and its performance is not as good as that of state-of-the-art methods

**Strengths:**

This manuscript attempts to construct a unified encoder for speech-related tasks in multi-talker scenarios. This research question is necessary and important.

**Weaknesses:**

- The manuscript lacks novelty. Numerous works have already discussed the use of multi-layer feature fusion/weighted sum, such as "Large-Scale Self-Supervised Speech Representation Learning for Automatic Speaker Verification," "Whisper-AT: Noise-Robust Automatic Speech Recognizers are Also Strong General Audio Event Taggers," and "Resource-Efficient Transfer Learning from Speech Foundation Model Using Hierarchical Feature Fusion." Compared to the aforementioned studies, this manuscript makes no novel contributions to methodology. Instead, it leverages existing techniques to address three speech-related tasks in multi-speaker scenarios.

- The authors overclaimed achieving state-of-the-art performance in speaker diarization, speech separation, and multi-speaker ASR.
  - However, for speech diarization, they only compared their method with ConvTasNet (2019) and did not benchmark against more recent methods such as Mossformer or Mossformer2.
  - For multi-speaker ASR, the proposed method achieved a WER of 9.2% on LibriSpeechMix. In contrast, the state-of-the-art performance is 3.43%, as reported in "Empowering Whisper as a Joint Multi-Talker and Target-Talker Speech Recognition System."
  - For speaker diarization, the authors only compared their method with EEND-based methods. The performance of their UME when combined with non-E2E method is not explored.

- The author reported the model's performance solely on two-speaker simulated overlapped speech for all three tasks. The performance in real-world scenarios and with more speakers remains under investigation.

**Questions:**

1. Can you explain the novelty of your approach compared to existing multi-layer weighted sum methods, aside from fine-tuning the model on the three multi-speaker-related tasks?
2. Why did you not compare your method with the latest research?
3. Why did you not report the results from other papers that demonstrate better performance?
4. It would be beneficial to discuss the performance on a broader range of datasets, rather than limiting the evaluation to only two-speaker LibriMix and LibriSpeechMix.

---

> ### Author Response · Authors · 2024-11-23
>
> **We thank the reviewer for the detailed feedback and constructive questions. We are encouraged by the recognition of the strengths of our work.**
>
> Below are the detailed responses to each query.
>
> **Weaknesses**
>
> >The manuscript lacks novelty. Numerous works have already discussed the use of multi-layer feature fusion/weighted sum, such as "Large-Scale Self-Supervised Speech Representation Learning for Automatic Speaker Verification," "Whisper-AT: Noise-Robust Automatic Speech Recognizers are Also Strong General Audio Event Taggers," and "Resource-Efficient Transfer Learning from Speech Foundation Model Using Hierarchical Feature Fusion." Compared to the aforementioned studies, this manuscript makes no novel contributions to methodology. Instead, it leverages existing techniques to address three speech-related tasks in multi-speaker scenarios.
>
> We acknowledge the reviewer’s comment regarding the lack of novelty in the multi-layer feature fusion/weighted sum approach. As noted in Section 2.1, our contribution lies in leveraging the weighted sum of OWSM encoder features to develop a unified speech model capable of addressing diarization, separation, and multi-speaker ASR tasks simultaneously. Integrating these tasks into a single multi-task framework presents a significant challenge due to their inherently disjoint nature. Our work addresses this issue by developing a unified framework that aligns these tasks through shared representations while allowing task-specific adaptations. This approach not only reduces redundancy in task-specific models but also promotes mutual benefits, such as improved speaker attributed transcription and better signal separation for diarization.
>
> ***
> >The authors overclaimed achieving state-of-the-art performance in speaker diarization, speech separation, and multi-speaker ASR. However, for speech diarization, they only compared their method with ConvTasNet (2019) and did not benchmark against more recent methods such as Mossformer or Mossformer2.
>
> Our main contribution lies in demonstrating the feasibility and flexibility of supporting end-to-end (E2E) speaker diarization, speech separation, and multi-speaker ASR tasks within a unified framework. To illustrate this, we adopt ConvTasNet, a widely recognized open-source speech separation model that operates in the time domain. Our experiments show that integrating ConvTasNet with speaker diarization and multi-speaker ASR tasks in a multi-task configuration improves its performance compared to training ConvTasNet as a standalone model. We believe this approach is not limited to ConvTasNet and could similarly enhance other speech separation models, such as Mosformer or Mosformer2, if they become available as open-source models.  This demonstrates the flexibility of our multi-task architecture and its potential to improve speech separation models when integrated with diarization and ASR tasks.
>
> ***
> >For multi-speaker ASR, the proposed method achieved a WER of 9.2% on LibriSpeechMix. In contrast, the state-of-the-art performance is 3.43%, as reported in "Empowering Whisper as a Joint Multi-Talker and Target-Talker Speech Recognition System."
>
> We consider this paper "contemporaneous" based on the reviewer guide provided by ICLR (https://iclr.cc/Conferences/2025/ReviewerGuide [FAQ for Reviewers]), which states:
>
> ```
> “Q: Are authors expected to cite and compare with very recent work? What about non-peer-reviewed (e.g., ArXiv) papers? (updated on 7 November 2022)
> A: We consider papers contemporaneous if they are published within the last four months. That means, since our full paper deadline is October 1, if a paper was published (i.e., at a peer-reviewed venue) on or after July 1, 2024, authors are not required to compare their work to that paper. Authors are encouraged to cite and discuss all relevant papers, but they may be excused for not knowing about papers not published in peer-reviewed conference proceedings or journals, which includes papers exclusively available on ArXiv. Reviewers are encouraged to use their own good judgment and, if in doubt, discuss with their area chair.”
> ```
>
> The non-peer-reviewed version of the referenced paper (https://arxiv.org/abs/2407.09817) was published on ArXiv on **July 13, 2024**, while the peer-reviewed version (https://www.isca-archive.org/interspeech_2024/meng24c_interspeech.html) was published on **September 1, 2024**. Given that our submission deadline was **October 1, 2024**, we did not include the results from this paper in our initial version as they fall within the "contemporaneous" period defined by the reviewer guide.

---

> > ### Author Response · Authors · 2024-11-23
> >
> > Additionally, as per the reviewer’s guide, a submission should not be rejected solely for not achieving state-of-the-art results. To quote:
> >
> > ```
> > “Q: If a submission does not achieve state-of-the-art results, is that grounds for rejection?
> > A: No, a lack of state-of-the-art results does not by itself constitute grounds for rejection. Submissions bring value to the ICLR community when they convincingly demonstrate new, relevant, impactful knowledge. Submissions can achieve this without achieving state-of-the-art results.”
> > ```
> >
> > While we respect the reviewers' feedback, we carefully reviewed the manuscript and decided to exclude the results reported by Meng et al. for the following reasons:
> >
> > * Meng et al.'s study used the "clean" subsets of Libri2Mix and Libri3Mix, whereas our work focuses on the more challenging noisy "mix both" subset (see Section 4.1).
> > * Including their results would introduce inconsistency in comparisons due to the differences in dataset characteristics and noise conditions.
> >
> > We hope this clarification addresses the reviewers' concerns.
> >
> > ***
> > >For speaker diarization, the authors only compared their method with EEND-based methods. The performance of their UME when combined with non-E2E method is not explored.
> >
> > Our main contribution lies in demonstrating the feasibility and flexibility of supporting end-to-end (E2E) speaker diarization, speech separation, and multi-speaker ASR tasks within a unified framework. To illustrate this, we adopt EEND, a widely recognized open-source end-to-end speaker diarization model. Our experiments show that integrating EEND with speech separation and multi-speaker ASR tasks in a multi-task configuration improves its performance compared to training EEND as a standalone model.
> >
> > ***
> > >The author reported the model's performance solely on two-speaker simulated overlapped speech for all three tasks. The performance in real-world scenarios and with more speakers remains under investigation.
> >
> > We will include results for Libri3Mix and LibriSpeech3Mix for completeness. However, due to the lack of ground truth data for all three tasks in publicly available real-world datasets, our analysis is limited to simulated datasets. This absence of ground truth in real-world data makes it challenging to objectively evaluate the UME framework's performance. Therefore, we focus on simulated datasets, where ground truth is available, to ensure accurate comparisons. We acknowledge this limitation and plan to explore real-world datasets in future work as they become more accessible and standardized.
> >
> > ***

---

> > > ### Comment · Reviewer_7DgA · 2024-12-02
> > >
> > > Thank you for the response. Some of my concerns (on performance) have been addressed, though I still think the manuscript falls below the acceptance standards for ICLR (especially for novelty).

---

### Official Review · Reviewer_xgGR · 2024-10-31

**Soundness:** 1
**Presentation:** 1
**Contribution:** 3
**Rating:** 3
**Confidence:** 2

**Summary:**

This work proposes a method where one speech foundation model is used as backbone to three task-specific heads. Unlike the standard SUPERB protocol, these heads are trained in a multi-task setup to perform speaker diarization, speech separation, and multi-speaker speech recognition.

**Strengths:**

This work tackles the inherently difficult problem of solving different, orthogonal speech technology tasks. A real-world ASR system needs to solve these, and currently this is done with a pipeline approach of multiple models. Unifying this pipeline of speech separation, speaker diarization, speech recognition, etc into a single model is beneficial to the community.

**Weaknesses:**

I do not believe this work uses a fair evaluation protocol, as it is stated in Section 4.1 that the the speaker diarization test set uses 100% overlapped speech. From my understanding of DER, this means that the model needs to simply predict that each of the 2 speakers are talking all the time. The low but not perfect scores in Table 1 are then simply due to noise in the ground-truth labels?

The paper is well-structured, but there are some sentences which are not clear or contain typos. See e.g.,
* line 37-38 "A key limitation of training tasks independently is that inter-dependencies cannot be leveraged"
* line 40 "Most existing speech-processing frameworks address this limitation with unified, joint training architectures" (also, 'unified, joint training architecture' is vague)
* line 48: delete " so much", add, e.g., "do not work well on"
* line 154 "gorund truth"
* line 156 "where a T-length..."
* line 298 "metrics" -> "metric"
* line 367 "Unlike WavLM ... outperforms WavLM" This sentence is hard to parse, split it up, e.g.: "WavLM uses overlapped speech mixtures, while OWSM is just trained on clean speech. We observe that... OWSM > WAVLM"
* line 369 "We explain" -> "We speculate" or "We hypothesize"

I believe reproduction of this work is difficult due to missing details on dataset generation and model training, see my questions below.

**Questions:**

* Can the authors comment on the use of 100% overlapped speech for evaluation of SD? Does the SUPERB benchmark for SD also use 100% overlapped speech?
* What model type was used for OWSMv3.1(base, small, medium , LR)? From Figure 3 I assume medium, but I do not think the text states this anywhere.
* What does a "flat start" mean in line 342?
* For how many epochs/steps did you train in total?
* How much VRAM is needed to train your model, which GPU(s) did you use, and how long did it take to train?
* What batch size did you use, and how were these sampled?
* What is the minimum, maximum, and average duration of utterances in your train/eval dataset(s)?
* I think it is unexpected (From Figure 3 and Figure 4) that the weights are nearly identical between task heads. Do the authors know what weights are found when training in a single-task setup, and whether they are similar or different to the values displayed in these figures?

---

> ### Author Response · Authors · 2024-11-23
>
> **We thank the reviewer for the detailed feedback and constructive questions. We are encouraged by the recognition of the strengths of our work.**
>
> Below are the detailed responses to each query.
>
> **Weaknesses**
> >The paper is well-structured, but there are some sentences which are not clear or contain typos.
>
> We have improved the overall clarity of the paper by refining the sentences and correcting the spelling mistakes indicated by the reviewer xgGR.
>
> **Questions**
> >Can the authors comment on the use of 100% overlapped speech for evaluation of SD? Does the SUPERB benchmark for SD also use 100% overlapped speech?
>
> We follow previous studies [1,2] to evaluate the speaker diarization task using 100% overlapped speech in the simulated dataset. Furthermore, to the best of our knowledge, the SUPERB benchmark, as reported by its authors [2], focuses on a two-speaker scenario in the LibriMix dataset, which also consists entirely of 100% overlapped speech.
>
> ***
> >What model type was used for OWSMv3.1(base, small, medium , LR)? From Figure 3 I assume medium, but I do not think the text states this anywhere.
>
> In this study, we employed "OWSMv3.1 medium," which will be explicitly mentioned in the manuscript.
>
> >What does a "flat start" mean in line 342? & For how many epochs/steps did you train in total?
>
> During training in UME, we initialize the encoder parameters with the pre-trained OWSMv3.1 medium encoder and fine-tune the encoder layers for 70 epochs, while all task-specific parameters have a flat start (i.e., no parameter initialization for task-specific layers) and are trained for 70 epochs. For the ASR-initialized UME model, the multi-speaker ASR model is pre-trained separately for 30 epochs, and then the ASR-specific head in the UME model is initialized from this pre-trained model. This results in a total of 70 epochs of fine-tuning for the OWSMv3.1 encoder layers, 70 epochs of training for the diarization and separation tasks, and 70 epochs of fine-tuning for the ASR task.
>
> We have added the above discussion in Section 4.3 of the revised manuscript.
> ***
> >How much VRAM is needed to train your model, which GPU(s) did you use, and how long did it take to train? & What batch size did you use, and how were these sampled?
>
> Four A100 80GB GPUs are used during training, and the batch size is dynamically adjusted based on the input length using the numel batch type in the ESPnet toolkit. In our experiments, the average batch size was 44, and it took six days to train the model for up to 70 epochs.
>
> We have added the above discussion in Section 4.3 of the revised manuscript.
> ***
> >What is the minimum, maximum, and average duration of utterances in your train/eval dataset(s)?
>
> The training and evaluation sets have utterance durations with a minimum of 3.1 seconds, a maximum of 56.8 seconds, and an average of 16.2 seconds. Additional details can be found in the corresponding table and are further discussed in Section 4.1 and Appendix A.2 of the revised manuscript.
> ***
> >I think it is unexpected (From Figure 3 and Figure 4) that the weights are nearly identical between task heads. Do the authors know what weights are found when training in a single-task setup, and whether they are similar or different to the values displayed in these figures?
>
> We have incorporated the weights from the single-task setups and observed that their weight distributions differ from those in the multi-task setups. However, the overall trend remains consistent: the initial and final layers contribute more in the single-task setups, except for the diarization task, where all layers contribute equally.
>
> We have added these observations in the Appendix A1 of the revised manuscript.
>
> ***
> **References**
> ```
> [1] Chen, Sanyuan et al. “WavLM: Large-Scale Self-Supervised Pre-Training for Full Stack Speech Processing.” IEEE Journal of Selected Topics in Signal Processing 16 (2021): 1505-1518.
> [2] Yang, Shu-Wen et al. “SUPERB: Speech processing Universal PERformance Benchmark.” Interspeech (2021).
> ```

---

> ### Comment · Reviewer_xgGR · 2024-11-26
>
> I want to thank the authors for their clear reply and updating the work in such a short time-span. I believe the results and conclusions as-is are not relevant to the community due to the 100% overlapping speech in the SD task, so I will not update my score.
>
> However, this also means that the SD task in SUPERB is not relevant, which I did not know before this review, and is something which I want to explicitly state is non-obvious and the authors should not blame themselves for having followed the SUPERB SD recipe.

---

> > ### Author Response · Authors · 2024-11-27
> > **Response to Reviewer xgGR**
> >
> > We appreciate the reviewer’s observation regarding the overlapping style of datasets like Libri2Mix and Libri3Mix, which are indeed 100% overlapped and left-aligned. This alignment simplifies tasks like speaker diarization, as it requires predicting speaker activity stamps only on the right side. However, we believe the evaluation remains fair since our proposed method and the baselines in the prior work [1,2] are evaluated under the same experimental conditions.
> >
> > That said, as shown in Table 3, left-aligned datasets pose significantly greater challenges for the multi-speaker ASR task than speaker diarization. This underscores the non-trivial nature of proposing a unified framework capable of handling tasks with inherently disjoint characteristics. As discussed in the paper, we address this complexity by directly leveraging the Libri2Mix and Libri3Mix datasets to demonstrate how our UME framework seamlessly integrates these diverse tasks. This approach reflects the framework’s ability to achieve broader objectives, such as task efficiency, generalizability, and scalability, which we believe are of particular interest to the ICLR audience.
> >
> > Additionally, to the best of our knowledge, our work makes a meaningful contribution as it is the first effort to unify speaker diarization, speech separation, and multi-speaker ASR in a single framework. This unification is achieved by leveraging a weighted sum of hidden state representations alongside multi-task learning, as detailed in Section 2.
> >
> > While we respect the reviewer’s perspective on the paper, we remain confident that our work represents a significant step toward addressing the broader challenges of unifying diverse tasks within end-to-end speech processing frameworks.

---

### Official Review · Reviewer_roJK · 2024-11-04

**Soundness:** 4
**Presentation:** 3
**Contribution:** 2
**Rating:** 5
**Confidence:** 5

**Summary:**

The authors introduce a method to concurrently train models for speech recognition, speech separation, and speaker diarization, utilizing embeddings from a pre-trained speech foundation model (SFM). They leverage a weighted sum of the model’s layer embeddings as input for each task. Experimental results indicate that this joint training approach enhances the performance of all three tasks.

**Strengths:**

The paper introduces a unified architecture that tackles Speech Separation, Speaker Diarization, and ASR tasks simultaneously. Experimental results confirm that this multitask framework enhances the performance of each task, demonstrating their mutual benefit.

**Weaknesses:**

As noted by the authors, previous works have utilized SFMs like WavLM for SS and ASR tasks. Although the proposed extension to cover all three tasks shows metric improvements compared to other SFMs, the innovation is minimal.

Additionally, the authors have only tested their approach on a simulated dataset. A more comprehensive evaluation using datasets like AMI, ICSI, or LibriCSS would better validate their method.

**Questions:**

1. Clarification on Figure 1:
Figure 1 appears misleading. Equation (3) shows different weights for different tasks, yet Figure 1 depicts only a single weight. Can you clarify this discrepancy?


2. Training with Real Data:
How do you propose to use real data for training the model, given that datasets for these tasks (e.g., ASR and speech separation) are typically disjoint? For example datasets for ASR do not generally contain ground truth for speech separation. How will this disjoint nature impact the joint training of the model?

---

> ### Author Response · Authors · 2024-11-23
>
> **We thank the reviewer for the detailed feedback and constructive questions. We are encouraged by the recognition of the strengths of our work.**
>
> Below are the detailed responses to each query.
>
> **Questions**
> >Clarification on Figure 1: Figure 1 appears misleading. Equation (3) shows different weights for different tasks, yet Figure 1 depicts only a single weight. Can you clarify this discrepancy?
>
> We have made the corrections in Figure 1.
> ***
> >Training with Real Data: How do you propose to use real data for training the model, given that datasets for these tasks (e.g., ASR and speech separation) are typically disjoint? For example datasets for ASR do not generally contain ground truth for speech separation. How will this disjoint nature impact the joint training of the model?
>
> Thank you for raising this important point regarding the challenge of training with real data for the joint tasks of separation, diarization, and ASR. We acknowledge that, in practice, datasets for these tasks are often disjoint, with ASR datasets typically lacking ground truth for separation and diarization, and vice versa. Here is our approach to mitigate this limitation:
>
> * We leverage synthetic or simulated datasets where ground truth for all three tasks (separation, diarization, and ASR) is available. This pre-training step enables the model to learn joint representations effectively and establish a foundational understanding of each task.
> * Once the model is pre-trained on synthetic data, we fine-tune it using real-world data for each task independently, while keeping the other task parameters frozen. This strategy will enable the professionals to use real diarization, separation and ASR datasets to adapt to each component of the UME framework.
>
> Similar strategies have been employed in existing studies to address the absence of ground truth for all three tasks in real-world datasets, e.g., as reported in [1] and [2]:
>
> **References**
> ```
> [1] Bredin, H. (2023) pyannote.audio 2.1 speaker diarization pipeline: principle, benchmark, and recipe. Proc. INTERSPEECH 2023, 1983-1987, doi: 10.21437/Interspeech.2023-105
> [2] Cornell, S., Wiesner, M.S., Watanabe, S., Raj, D., Chang, X., Garcia, P., Masuyam, Y., Wang, Z.-Q., Squartini, S., Khudanpur, S. (2023) The CHiME-7 DASR Challenge: Distant Meeting Transcription with Multiple Devices in Diverse Scenarios. Proc. 7th International Workshop on Speech Processing in Everyday Environments (CHiME 2023), 1-6, doi: 10.21437/CHiME.2023-1
> ```

---

> > ### Comment · Reviewer_roJK · 2024-11-25
> > **Thank you for addressing my questions**
> >
> > I thank the authors of the paper for addressing my questions and updating the paper. However the concerns raised regarding the weakness of the paper still remains and I will retain my rating for the paper.

---

> > > ### Author Response · Authors · 2024-11-27
> > > **Response to Reviewer roJK**
> > >
> > > We respect the reviewer’s feedback; however, our experiments are limited to simulated datasets due to the lack of ground truth data for training all three tasks jointly for publicly available real-world datasets. We acknowledge this limitation in the section on limitations and plan to explore real-world datasets in future work as they become more accessible and standardized.

---

### Author Response · Authors · 2024-11-23
**Summary of changes in the revised manuscript**

**Summary of changes in the revised manuscript**

We have revised the manuscript and uploaded an updated version incorporating the following changes:

* Corrected the variable label in figure 1.
* Added the minimum, maximum, and average durations of utterances in Section 4.1.
* Clarified details about the implementation, experimental setup, and the choice of weights for task-specific losses.
* Included Libri3Mix results for completeness in Tables 2, 3, and 4.
* Added a discussion of the Libri3Mix experiments in Section 5.2.
* Added recovered speech examples for the three-speaker case in Appendix A.3.
* Provided layer weights for single-task models in the Appendix A.1 and moved the previous layer weights figure there due to space constraints.
* Added a discussion on the limitations of using simulated datasets in the Limitations section.

**Note:** All updates in the revised paper are highlighted in red text for easy reference.

---

### Meta-Review · Area_Chair_8Vmg · 2024-12-08

**Metareview:**

This paper Introduces a unified architecture that simultaneously tackles Speech Separation, Speaker Diarization, and ASR tasks, enhancing the performance of each task through a multitask framework. It addresses the practical need for a unified model in real-world ASR systems, which currently relies on a pipeline approach with multiple models. It demonstrates that solving different, orthogonal speech technology tasks together can be mutually beneficial, improving overall system performance. The paper is clearly written and easy to follow, making the complex concepts accessible to readers.

However,  the manuscript leverages existing techniques without making novel contributions to methodology, similar to previous works on multi-layer feature fusion and joint training. The proposed extension to cover all three tasks (Speech Separation, Speaker Diarization, and ASR) shows metric improvements but lacks significant innovation compared to previous works. The approach is only tested on a simulated dataset, lacking comprehensive evaluation on real-world datasets like AMI, ICSI, or LibriCSS. The authors overclaim SOTA performance without adequate benchmarking against more recent methods for speaker diarization, speech separation, and multi-speaker ASR. The model's performance is only reported for two-speaker simulated overlapped speech, with real-world scenarios and more speakers remaining untested.

**Additional Comments On Reviewer Discussion:**

Although the authors addressed the reviewers' concerns, the reviewers felt that their concerns were not fully resolved and maintained their original scores.

---

### Decision · Program_Chairs · 2025-01-22

Reject